

**A Nonlinear Generalized Boussinesq Equation ((2+1)-D) for Rossby-Khantadze Waves**
**Laila Zafar Kahlon[1]*, Tamaz David Kaladze[2, 3], Hassan Amir Shah[1], Taimoor Zaka[1],**
**Syed Assad Ul Azeem Bukhari[1]**
[1]Physics Department, Forman Christian College (A Chartered University), Lahore 54600,
Pakistan
[2]I. Vekua Institute of Applied Mathematics, Tbilisi State University, 2 University str, Tbilisi
0186, Georgia
[3]E. Andronikashvili Institute of Physics, I. Javakhishvili Tbilisi State University, Tbilisi
0128, Georgia
*Corresponding author: Email address: lailakahlon@fccollege.edu.pk (Laila Zafar Kahlon)
**Abstract**
In the following paper, we investigate nonlinear Rossby-Khantadze waves at a higher
dimension, by taking the inhomogenities in the geomagnetic field and in angular velocity into
account. Considering the system to be weakly nonlinear, we make use of perturbation theory
to derive a new (2+1)–D general form of Boussineq equation, derived from the equation of
potential vorticity. We evaluate the obtained equation by using the qualitative theory of ODEs,
and bifurcation theory of dynamical systems. Through which we obtain the exact solution of
the system in a co-moving frame of reference and for more information, we make use of
dynamical analysis. Furthermore, we provide the exact numerical solutions. These results show
that the aforementioned solutions of the traveling waves corresponds to Rossby-Khantadze
solitons.

**Keywords:** Generalized Boussinesq model equation; nonlinear Rossby-Khantadze waves;
nonlinearity; sheared zonal flow; traveling wave solutions; dynamical analysis

**1. Introduction**

Numerous investigations conducted by ground-based and satellite observations gives
proof of the Zonal flow's existence in atmospheric regions of atmosphere (Pedlosky, 1987).
This is based on the fact of the non-uniform heating caused by the sun in the Earth's
atmospheric regions. These ULF perturbations in ionosphere E and F regions occurr due to the
sheared flow with nonhomogeneous velocities along the meridians (Shukla et al., 2003;
Onishchenko et al. 2004; Satoh, 2004; Kaladze et al., 2007; Kaladze et al., 2008). The sheared
flow affects properties of such linear and nonlinear waves in the ionosphere. Under certain
suitable conditions they give rise various nonlinear structures like zonal flows (ZFs), vortices,
solitons etc.
Sheared Rossby waves have gained much attention due to their prominent role in the
global atmospheric circulation. It must be noted that the spatial inhomogeneity, along the
meridians, of both the background field (magnetic) and the force (Coriolis) parameter makes
such coupled modes, called the Rossby-Khantadze (RK) propagation (see e.g. Kaladze et al.
2011). It is discussed that sheared RK electromagnetic vortices in the E ionospheric region
(Kaladze et al., 2011; 2012; 2013a, 2013b; 2014a, 2014b). In the aforementioned papers, the
self-organization of coupled RK waves into solitary dipolar vortices alongwith the possibility
of the intensive magnetic field is shown. In the recent work, different nonlinear processes
having relevance to the generation of zonal flows (sheared) by Rossby waves are considered.



The key factor for the generation of zonal flows in short-wavelength Rossby waves is Reynold's stress (Shukla et al., 2003 and Onishchenko et al. 2004). The Rossby waves causes the generation of zonal flows in E ionosphere was investigated by Kaladze et al. (2007). Such nonlinear Rossby wave structures are splitted into various parts having dependent on zonal flow'' energy (Kaladze et al., 2008). Along with the analytical side, numerical work of RK waves with sheared zonal flows in the E layer of the ionosphere is worked out as well (Futatani et al., 2013, 2015). In these work, breaking of vortices is studied where the energy is transfered from sheared flow into these multiple pieces. While the equatorially propagating Rossby solitary waves by sheared flows have also been discussed (Qiang et al., 2001) and the presence of such solitary structures was confirmed by *Freja and Viking satellites* in work of Bostrom, 1992; Lindqvist et al., 1994; Dovner et al. 1994; Qiang et al., 2001). In Jian et al., (2009)'s work, the authors studied the nonlinear propagation of sheared Rossby waves in stratified neutral fluids and obtained modified Korteweg-de Vries (MKdV) equation, which is characterised by a cubic nonlinearity. Kahlon et al. (2024) investigated the MKdV equation with cubic nonlinearity for Rossby-Khantadze nonlinear waves.

Zonal flow's generation in the ionosphere's E region by Rossby-Khantadze waves having magnetic field have also been shown (Kaladze et al. 2012, Kahlon and Kaladze 2015). It has been predicted that there exists a possibility of the magnetic field generation, at the strength of $10^3 \, nT$. Kaladze et al. (2019) studied the nonlinear interaction of magnetized Rossby waves with inclusion of zonal flows in the Earth's ionospheric E-layer, in which they obtained MKdV solitons. The possibility of planetary Rossby wave's existence in the dynamo E-area of weakly ionised ionosphere was predicted by Forbes, 1996. It was also shown to correspond with the experimental interpretations. Much later, Vukcevic and Popovic, (2020) investigated the possibility of soliton formation at different latitudes in ionosphere. Direct observed data of satellites of such soliton structures from Earth'surface are discussed.

In the context of shallow water waves and in plasmas, several researchers have extended the KdV and MKdV equations to higher dimensions, in order to obtain realistically accurate results. Notably, Kadomstev-Petviashvilli (KP) equation and Zakharov-Kuznetsov (ZK) equation have gained much attention (Vukcevic et al., 2020, Kadomstev et al., 1970, Groves et al., 2008, Infeld et al., 2000 and Zakharov et al., 1974). Both of those equations are (2+1) – dimensional in nature, and are very useful in plasma models (as one can get almost complete information by taking parallel and perpendicular dimension into account). While modelling shallow water waves, Johnson (1996) investigated a (2+1) – dimensional Boussinesq equation to studied gravitational surface waves. Making use of the surface wave theory, Mitsotakis (2009) investigated the Boussinesq equation and simulated the propagation of such waves. In the context of geophysics, many authors (Gottwald, 2003, Yang et al., 2016, Yang et al., 2018, Zhang et al., 2017, Zhang et al., 2017) have investigated ZK equation by considering nonlinear Rossby waves from the quasi-geostropic potential vorticity equation. Although, the Boussinesq equation in the study of the nonlinear Rossby-Khantadze waves is not reported so far.

It is very useful to find exact and the explicit nonlinear solutions of partial differential equations (NLPDEs). Recently, several techniques have been used to find such solutions, including but not limited to the method of trigonometric series (Ma and Fuchssteiner, 1996), the method of tan($\phi(\xi)/2$)-expansion (Manafian and Aghdaini, 2016), the sine-cosine method (Wazwaz, 2005), the Wronskian method (Ma and You, 2005), separation of variables approach (Lin and Zhang, 2007), the Septic B-spline method (El-Danaf, 2008), the transformative functional rational method (Ma and Lee, 2009), the symmetry algebra method (Ma and Chen, 2009) the mesh-free method (Haq and Uddin, 2009), the homotopy perturbation method (Ganji et al., 2009), the modified mapping method and the extended mapping method (Zhang et al., 2010), qualitative theory of the bifurcation method and dynamical systems (Zhang et al., 2011),



the multiple exp-function method (Ma and Zhu, 2012), the modified (G'/G)- method of
expansion (Miao and Zhang, 2011), the modified trigonometric function series method (Zhang
et al 2011) infinite series method and Jacobi elliptic functional method (Zhang et al., 2012,
Tasbozan et al., 2016), RBF approximation method (Uddin, 2014) (G' /G−1/G)-expansion
method (Zhang et al., 2014), Hirota bilinear method (Lu et al., 2016, Ma et al., 1996, 2016, Lu
and Ma, 2016), lattice Boltzmann method (Wang and Yan, 2016) to have some of the
techniques.
In the present work, for the weakly ionized and conducting ionosphere E plasma we
consider the stream-function and evolution of geomagnetic field for RK electromagnetic
waves, which provides novelty to this work. . In Sec. 2, we set the initial system of equations.
In Se. 3, by using the reductive perturbation technique we obtain the linear dispersion equation
from the lowest order of $\varepsilon$. In Sec. 4, we derive the Boussinesq equation for Rossby-Khantadze
nonlinear waves from our considered set of equations. In Sec. 5, we study the dynamical
analysis of the Boussinesq equation and get its exact traveling wave solutions. In last section,
discussions are presented in Sec. 6.
**2. Mathematical Preliminaries**
We start by considering a weakly ionised system, as is characteristic to ionospheric
plasmas. Here ions, electrons and neutral particles are embedded in a nonhomogeneous
geomagnetic field ergo, $\boldsymbol{B}_0(y) = \left(0, B_{0y}(y), B_{0z}(y)\right)$, and the angular velocity is taken into
consideration as, $\boldsymbol{\Omega}(y) = \left(0, \Omega_{0y}(y), \Omega_{0z}(y)\right)$. We consider the 2D incompressible motion i.e.,
$\mathbf{v} = (u, \mathrm{v}, 0)$, which represents the velocity of the neutral gas where $u = -\frac{\partial \psi}{\partial y}$, $\mathrm{v} = \frac{\partial \psi}{\partial x}$ and
$\psi(x, y, t)$ is the stream function.
We make use of a slab geometry with zonally x, latitudinally y, and locally vertical
direction z direction. Furthermore the behavior of the nonlinear Rossby-Khantadze sheared
electromagnetic waves could be expressed by the 2D system of equations (e.g. Kaladze et al.,
2011, Kaladze et al., 2014, Song et al. 2009; Liu et al. 2019) given below:

$$\begin{cases} \frac{\partial \Delta \psi}{\partial t} + \beta \frac{\partial \psi}{\partial x} + \mathrm{J}(\psi, \Delta \psi) - \frac{1}{\mu_0 \rho} \beta_B \frac{\partial h}{\partial x} = -\mu \, \Delta \psi + Q \; , & (1a) \\ \frac{\partial h}{\partial t} + \mathrm{J}(\psi, h) + \beta_B \frac{\partial \psi}{\partial x} + c_B \frac{\partial h}{\partial x} = 0 \, , & (1b) \end{cases}$$

Here in the equation (1a) we consider vorticity as $\zeta_z = \boldsymbol{e}_z \cdot \nabla \times \mathbf{v} = \Delta \psi = \nabla^2 \psi = (\partial_x^2 +$
$\partial_y^2) \psi$ from momentum equation of single fluid where $\beta = \frac{\partial f}{\partial y} = \frac{2 \partial \Omega_{0z}}{\partial y}$ is the latitudinally
inhomogeneous angular velocity with $f = f_0 + \beta(y)y$ with $f_0 = 2\Omega_{0z} = 2\Omega_0 \sin \phi_0$. While
the parameter $c_B = \beta_B / en\mu_0$ with $\beta_B = \frac{\partial B_{0z}}{\partial y}$, is the nonhomogeneity in the geo-magnetic
field, $n$ is charged particles's number density, $J(a, b) = \frac{\partial a}{\partial x} \frac{\partial b}{\partial y} - \frac{\partial a}{\partial y} \frac{\partial b}{\partial x}$ is the Jacobian. The
equation (1b) shows the z-component of perturbed magnetic field. Note that lesser contribution
of charged particles (in comparison of neutrals) provides role (Kaladze, et al. 2013a, 2013b) in
the inductive current.
To solve the set of equation (1), we use the boundary condition

$$\left. \frac{\partial \psi}{\partial x} \right|_{y=y_1} = \left. \frac{\partial \psi}{\partial x} \right|_{y=y_2} = 0 \, , \qquad (2)$$






representing the flow along the merdional directions (Pedlosky (1987); Satoh (2004)).

By introducing the following dimensionless parameters, we can express Eq (1) in
dimensionless form
$$(x, y) = L_\circ(x^*, y^*), \quad \psi = L_0 U_0 \psi^*, \quad t = \frac{L_\circ}{U_\circ} t^*, \quad \beta = \frac{U_0}{L_0^2} \beta^*, \quad \mu = \frac{U_0}{L_0} \mu^*, \quad Q = \frac{U_0^2}{L_0^2} Q^* \quad (3)$$
Here asterisk denotes the dimensional variables, which are further dropped in the equation
below. Here $L_0$ is the zonally length; H is a vertically length and $U_0$ is the velocity. Finally,
Eq. (1) takes the form

$$\begin{cases} \frac{\partial \Delta \psi}{\partial t} + \beta \frac{\partial \psi}{\partial x} + J(\psi, \Delta \psi) - \frac{1}{\mu_0 \rho} \beta_B \frac{\partial h}{\partial x} = -\mu \, \Delta \psi + Q \ , \\ \frac{\partial h}{\partial t} + J(\psi, h) + \beta_B \frac{\partial \psi}{\partial x} + c_B \frac{\partial h}{\partial x} = 0 \ , \end{cases} \quad (4)$$


with the following boundary conditions
$$\left. \frac{\partial \psi}{\partial x} \right|_0 = \left. \frac{\partial \psi}{\partial x} \right|_1 = 0 \ . \quad (5)$$

**3. Perturbation and weakly nonlinear approach**
In this section, to investigate the non-linear Boussinesq equation describing the solitary
Rossby-Khantadze waves, we will use the multiple scale and asymptotic expansion approach.
The stream function is taken as
$$\psi = \bar{\psi} (y) + \psi'(x, y, t), \quad (6)$$
with $\bar{\psi} = - \int_0^y [\bar{u} (s) - c_0] \, ds$ represents the background stream function where $c_0$ is a
constant, $\bar{u}(y)$ refers to background flow, and $\psi'$ is the disturbance in stream function. While
the perturbed magnetic field is:
$$h = \varepsilon h', \quad (7)$$
Thus, the set of equations (4) can be expressed as

$$\begin{cases} \left( \frac{\partial}{\partial t} + (\bar{u} - c_0) \ \frac{\partial}{\partial t} \right) \Delta \ \psi' + p(y) \frac{\partial \psi'}{\partial t} + J(\psi', \Delta \psi') - \frac{\beta_B}{\mu_0 \, \rho} \frac{\partial h'}{\partial x} = -\mu \Delta^2 \psi' \\ \frac{\partial h'}{\partial t} + \varepsilon J(\psi', h') + (U(y) - c_0) \frac{\partial h'}{\partial x} + \beta_B \frac{\partial \psi'}{\partial x} + c_B \frac{\partial h'}{\partial x} = 0. \end{cases} \quad (8)$$

where $p(y) = (\beta(y)y - \bar{u}')'$ .
By applying the multiple scale approach,
$$X = \varepsilon^{(1/2)} x, \quad Y = \varepsilon(y - c_1 t) \quad T = \varepsilon \, t, \quad (9)$$





in the comoving frame of reference the differential operator can be expressed in the following
manner

$$\frac{\partial}{\partial x} = \varepsilon^{(1/2)} \frac{\partial}{\partial X}, \quad \frac{\partial}{\partial y} = \frac{\partial}{\partial y} + \varepsilon \frac{\partial}{\partial Y}, \frac{\partial}{\partial t} = \varepsilon \frac{\partial}{\partial T} - c_1 \varepsilon \frac{\partial}{\partial Y}. \tag{10}$$

The perturbed stream function and perturbed magnetic fields are expanded as

$$\begin{cases} \psi' = \varepsilon \psi_1 + \varepsilon^{(3/2)} \psi_2 + \varepsilon^2 \psi_3 + \cdots, \\ h' = \varepsilon h_1 + \varepsilon^{(3/2)} h_2 + \varepsilon^2 h_3 + \cdots. \end{cases} \tag{11}$$

Using (9), (10) and (11) into equation (7) we get from the lowest order i.e. $O(\varepsilon^{3/2})$:

$$\begin{cases} (\bar{u} - c_0) \frac{\partial}{\partial X} \left( \frac{\partial^2 \psi_1}{\partial y^2} \right) + p(y) \frac{\partial \psi_1}{\partial X} - \frac{\beta_B}{\mu_0 \rho} \frac{\partial}{\partial X}(h_1) = 0, \\ (\bar{u} - c_0 + c_B) \frac{\partial h_1}{\partial X} + \beta_B \frac{\partial}{\partial X}(\psi_1) = 0, \end{cases} \tag{12}$$

Next order $O(\varepsilon^2)$ gives

$$\begin{cases} (\bar{u} - c_0) \frac{\partial}{\partial X} \left( \frac{\partial^2 \psi_2}{\partial y^2} \right) + p(y) \frac{\partial \psi_2}{\partial X} = -\frac{\beta_B}{\mu_0 \rho} \frac{\partial}{\partial X}(h_1) - \left( \frac{\partial}{\partial T} - c_1 \frac{\partial}{\partial Y} \right) \frac{\partial^2 \psi_1}{\partial y^2}, \\ \left( \frac{\partial}{\partial T} - c_1 \frac{\partial}{\partial Y} \right) h_1 + (\bar{u} - c_0 + c_B) \frac{\partial h_2}{\partial X} + \beta_B \frac{\partial \psi_2}{\partial X} \end{cases} \tag{13}$$

From the second set of equation (13), we get

$$\frac{\partial h_2}{\partial X} = \frac{-\beta_B}{\bar{u} - c_0 + c_B} \frac{\partial \psi_2}{\partial X} - \frac{1}{(\bar{u} - c_0 + c_B)} \left( \frac{\partial}{\partial T} - c_1 \frac{\partial}{\partial Y} \right) h_1 \tag{14}$$


Next order $O(\varepsilon^{5/2})$ gives

$$\begin{cases} (\bar{u} - c_0) \frac{\partial}{\partial X} \left( \frac{\partial^2 \psi_2}{\partial y^2} \right) + p(y) \frac{\partial \psi_3}{\partial X} - \frac{\beta_B}{\mu_0 \rho} \frac{\partial h_3}{\partial X} = -(\bar{u} - c_0) \frac{\partial^3 \psi_1}{\partial X^3} - 2(\bar{u} - c_0) \frac{\partial^3 \psi_1}{\partial X \partial Y \partial y} \\ \qquad - \left( \frac{\partial}{\partial T} - c_1 \frac{\partial}{\partial Y} \right) \frac{\partial^2 \psi_2}{\partial y^2} - \frac{\partial \psi_1}{\partial X} \frac{\partial^3 \psi_1}{\partial y^3} + \frac{\partial \psi_1}{\partial y} \frac{\partial}{\partial X} \left( \frac{\partial^2 \psi_1}{\partial y^2} \right), \\ \left( \frac{\partial}{\partial T} - c_1 \frac{\partial}{\partial Y} \right) h_2 + \beta_B \frac{\partial \psi_3}{\partial X} = (\bar{u} - c_0 + c_B) \frac{\partial h_3}{\partial X} + \frac{\partial \psi_1}{\partial X} \frac{\partial h_1}{\partial Y} - \frac{\partial \psi_1}{\partial Y} \frac{\partial h_1}{\partial X}. \end{cases}$$

$$\tag{15}$$
Equation (15b) gives

$$\frac{\partial h_3}{\partial X} = - \left( \frac{\partial}{\partial T} - c_1 \frac{\partial}{\partial Y} \right) h_2 + \beta_B \frac{\partial \psi_3}{\partial X} + (\bar{u} - c_0 + c_B) + \frac{\partial \psi_1}{\partial X} \frac{\partial h_1}{\partial Y} - \frac{\partial \psi_1}{\partial Y} \frac{\partial h_1}{\partial X} \tag{16}$$


Assume that Eq. (12) has the solution
$$\psi_1 = A(X, Y, T) \varphi_1(y), \tag{17}$$

Thus, from equations (12) and (20) we get the following linear dispersion relation



$$\varphi_1'' \; + \; \frac{p(y)}{(\bar{u}-c_0)}\varphi_1(y) + \frac{\beta_B^2}{\mu_0\,\rho}\frac{1}{(\bar{u}-c_0)(\bar{u}-c_0+c_B)}\varphi_1 = 0, \tag{18}$$

and from the boundary condition given by Eq. (5) we get
$$\varphi_1(0) = \varphi_1(1) = 0. \tag{19}$$


The obtained Eq. (18) is the Rayleigh-Kuo equation describing the Rossby-Khantadze waves.
By solving Eq. (12) simultaneously and the coefficients are locally constant and $U(y)=const.$,
we get the following dispersion equation
$$\left(\left(\frac{\omega}{k_x}-U(y)\right)k_\perp^2 + p(y)\right)\left(\frac{\omega}{k_x}-U(y)-c_B\right)-\alpha = 0, \tag{20}$$

where $k_\perp^2 = k_x^2 + k_y^2$ and $\alpha = \frac{\beta_\beta^2}{\mu_0\rho}$. Eq. (20) describes the dispersion equation of sheared Rossby-
Khantadze waves. In the absence of $\alpha$ we get two solutions one independent solution of Rossby
waves and the second one for Khantadze waves.
By introducing the dimensionless variables $\frac{\omega}{k_x d}\Rightarrow v_p$ and $\frac{k_\perp^2 d}{a}\Rightarrow k_\perp^2$ (with $d=\frac{b}{en\mu_0}$, $a=\frac{2\Omega_0}{R}$
and $b=\frac{2b_{eq}}{R}$ ) then we rewrite the dispersion relation (20)
$$v_p = -U + \frac{1}{2k_\perp^2}\cos\lambda_0\left(-k_\perp^2 - 1 \pm \sqrt{(1-k_\perp^2)^2 + k_\perp^4\alpha_0}\right). \tag{21}$$

Here $\alpha_0 = \frac{ben}{ap} = \frac{x}{|c_B|\beta}$. For the E-ionosphere layer, the parameters have the following $B_{eq}\cong$
$0.5\times 10^{-4}$T, $2\Omega_0 \cong 10^{-4}\frac{rad}{s}, \frac{n}{N}\sim 10^{-8}$- $10^{-6}$, $\rho = (10^{-7}$ -$10^{-8})$ kg$m^{-3}$, the parameter $\alpha_0 =$
$(10^{-2}-1)$ .[Kaladze et al., 2011]
In Fig. 1, the phase velocity $v_p$ of coupled Rossby-Khantadze waves is plotted with
wave number $k_\perp$ by varying $\alpha_0$. Red curve is for "+" and blue is for "–" signs before the
radicaand in Eq. (21).

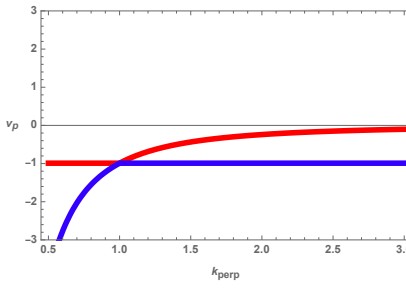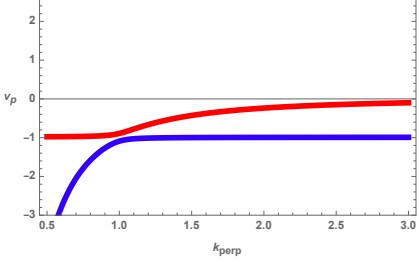


a) $\alpha_0 = 0$                    b) $\alpha_0 = 0.01$









$$c) \quad \alpha_0 = 0.1 \qquad\qquad\qquad d) \quad \alpha_0 = 1$$



$$e) \; \alpha_0 = 2$$

Fig.1 The phase velocity $v_p$ vs wave number $k_\perp$ of coupled Rossby-Khantadze waves for
$$\lambda_0 = \pi/4 \, .$$
**4 Derivation for the nonlinear Boussinesq Equation**

In this section, by taking into account the separation of variables techniques we

will derive the nonlinear Boussinesq Equation describing the solitary nonlinear structures.

Further, we assume that equation (13) has the solution

$$\psi_2 \;=\; \psi_{21} \;+\; \psi_{22}, \tag{22}$$
with
$$\psi_{21} = B_1(X,Y,T)\,\varphi_{21}(y)\,, \quad \psi_{22} = B_2(X,Y,T)\,\varphi_{22}(y), \tag{23}$$

By using the separation of variables approach, we obtain from (13) by using (22) and (23)

$$(\bar{u}-c_0)\frac{\partial B_2}{\partial X}\,\varphi''_{22} + \left(p(y)+\frac{\beta_B^2}{\mu_0\,\rho\,(c_B+\bar{u}-c_0)}\right)\frac{\partial B_2}{\partial X} = c_1\frac{\partial A}{\partial y}\,\varphi''_1 - \frac{\alpha c_1}{(c_B+\bar{u}-c_0)^2}\frac{\partial A}{\partial Y}\varphi_1 \tag{24}$$
Put





$$\frac{\partial B_1}{\partial X} = \frac{\partial A}{\partial T}, \qquad \text{and} \qquad \frac{\partial B_2}{\partial X} = \frac{\partial A}{\partial Y}. \qquad (25)$$
From Eq. (24) we get
$$\varphi_{21}'' + q(y)\varphi_{21} = -\frac{\varphi_1''}{\bar{u} - c_o} + \gamma\varphi_1 \qquad (26)$$
$$\varphi_{21}(0) = \varphi_{21}(1) = 0. \qquad (27)$$
with $q(y)$ and $\gamma$ are given by $q(y) = \left. \left( p(y) + \frac{\beta_B^2}{\mu_0 \rho} \cdot \frac{1}{(\bar{u} - c_0)(u - c_0 + c_B)} \right) \middle/ (\bar{u} - c_o) \right.; \gamma =$
$\frac{\beta_B^2}{\mu_0 \rho} \frac{1}{(\bar{u} - c_0)(\bar{u} - c_0 + c_B)^2}$.
And
$$\varphi_{22}'' + q(y)\varphi_{22} = \frac{c_1\varphi_1''}{\bar{u} - c_o} - c_1\gamma_1\varphi_1, \qquad (28)$$
The boundary conditions are given by
$$\varphi_{22}(0) = \varphi_{22}(1) = 0. \qquad (29)$$
From Eqs. (26) and (28) we have
$$\varphi_{22} = -c_1 \varphi_{21} \qquad (30)$$
In order to arrive at the evolution equation we use Eqs. (20), (25) and (26) and substitute into
Eq. (15)

$$(\bar{u} - c_0)\frac{\partial}{\partial X}\left(\frac{\partial^2 \Psi_3}{\partial y^2}\right) + p(y)\frac{\partial \Psi_3}{\partial X} = F, \qquad (31)$$
where
$F = -\varphi_{21}'' \frac{\partial^2 B_1}{\partial T \partial X} - \varphi_{22}'' \frac{\partial^2 B_2}{\partial T \partial X} + c_1\varphi_{21}'' \frac{\partial^2 B_1}{\partial Y \partial X} + c_1\varphi_{21}'' \frac{\partial^2 B_2}{\partial Y \partial X}(1 + \frac{\alpha}{(c_B + \bar{u} - c_0)^2} - (\bar{u} - c_0)\varphi_1 \frac{\partial^4 A}{\partial X^4} -$
$2(\bar{u} - c_0)\varphi_1 \frac{\partial^3 A}{\partial X^2 Y} - (\varphi_1\varphi_1''' - \varphi_1'\varphi_1'')2A\frac{\partial^2 A}{\partial X^2} + \frac{\alpha}{(c_B + \bar{u} - c_0)^3}(\frac{\partial^2}{\partial T^2} - 2c_1 \frac{\partial^2}{\partial T \partial Y} + c_1^2 \frac{\partial^2}{\partial Y^2})A\varphi_1$
$$(32)$$

Eq. (31) is the evolution equation for $\Psi_3$ and we obtain its solution by multiplying by $\varphi_1(y)$
and then integrating over $y$ to get

$\int_0^1 \frac{\varphi_1(y)}{\bar{u} - c_0}- [- \varphi_{21}'' \frac{\partial^2 B_1}{\partial T \partial X} -\varphi_{22}'' \frac{\partial^2 B_2}{\partial T \partial X} +c_1\varphi_{21}'' \frac{\partial^2 B_1}{\partial Y \partial X} +c_1\varphi_{21}'' \frac{\partial^2 B_2}{\partial Y \partial X}(1 + \frac{\alpha}{(c_B + \bar{u} - c_0)^2} - (\bar{u} - c_0)\varphi_1 \frac{\partial^4 A}{\partial X^4} -$
$2((\bar{u} - c_0)\varphi_1 \frac{\partial^3 A}{\partial X^2 Y} - (\varphi_1\varphi_1''' -\varphi_1'\varphi_1'') 2A\frac{\partial^2 A}{\partial X^2} +\frac{\alpha}{(c_B + \bar{u} - c_0)^3}(\frac{\partial^2}{\partial T^2} -2c_1 \frac{\partial^2}{\partial T \partial Y} +c_1^2 \frac{\partial^2}{\partial Y^2})A\varphi_1]$ dy
$$(33)$$





$\quad I_1 \frac{\partial^2 B_1}{\partial X \partial T} + I_2 \frac{\partial^2 B_2}{\partial X \partial T} - c_1 I_1 \frac{\partial^2 B_1}{\partial X \partial Y} - c_1 I_2 \frac{\partial^2 B_2}{\partial X \partial Y} + I_3 \frac{\partial^4 A}{\partial X^4} + I_4 \frac{\partial^3 A}{\partial X^2 \partial Y} + I_5 A \frac{\partial^2 A}{\partial X^2} + I_6 \left( \frac{\partial^2 A}{\partial T^2} - 2c_1 \frac{\partial^2 A}{\partial T \partial Y} + c_1^2 \frac{\partial^2 A}{\partial Y^2} \right) = 0$

(34)

where the coefficients are :

$$
\begin{cases}
I_1 = \int_0^1 \frac{q(y)\varphi_1}{\bar{u} - c_0} \left( \varphi_{21} - \left( 1 + \frac{\gamma(\bar{u} - c_0)}{q(y)} \right) \frac{\varphi_1}{\bar{u} - c_0} \right) \left( 1 + \frac{\alpha}{(c_B + \bar{u} - c_0)^2} \right) dy \; ; \\[3mm]
I_2 = \int_0^1 \frac{q(y)\,\varphi_1}{\bar{u} - c_0} \left( \varphi_{22} + \left( 1 + \frac{\gamma(\bar{u} - c_0)}{q} \right) \frac{\varphi_{1(y)}}{\bar{u} - c_0} \right) \left( 1 + \frac{\alpha}{(c_B + \bar{u} - c_0)^2} \right) dy; \\[3mm]
I_2 = c_1 I_1 = 2c_1 \int_0^1 \frac{q(y)\varphi_1^2}{(\bar{u} - c_0)^2} \left( 1 + \frac{\gamma(\bar{u} - c_0)}{q} \left( 1 + \frac{\alpha}{(\bar{u} - c_0 + c_B)^2} \right) \right) dy; \\[3mm]
I_3 = - \int_0^1 \varphi_1^2 \, dy \; ; \\[3mm]
I_4 = -2 \int_0^1 \varphi_1 \, \varphi_1' dy \; ; \\[3mm]
I_5 = \int_0^1 \frac{\varphi_1^3 \, q'}{\bar{u} - c_0} \, dy \; ; \\[3mm]
I_6 = \int_0^1 \left( \frac{\partial^2 A}{\partial T^2} - 2c_1 \frac{\partial^2 A}{\partial T \partial Y} + c_1^2 \frac{\partial^2 A}{\partial Y^2} \right).
\end{cases}
$$


(35)

Noting that

$$
\frac{\partial^2 B_1}{\partial X \partial T} = \frac{\partial^2 A}{\partial T^2} \; ; \; \frac{\partial^2 B_2}{\partial X \partial T} = \frac{\partial^2 A}{\partial Y \partial T} \quad \text{as} \quad \frac{\partial B_1}{\partial X} = \frac{\partial A}{\partial T} \; ; \quad \frac{\partial B_2}{\partial X} = \frac{\partial A}{\partial Y} \tag{36}
$$

By using (36) in Eq. (34) which gives

$$
\frac{\partial^2 A}{\partial T^2} + \left( \frac{(I_2 - 2c_1 I_6 - c_1 I_1)}{I_1 + I_6} \right) \frac{\partial^2 A}{\partial Y \partial T} - \left( \frac{I_6 c_1^2 - c_1 I_2}{I_1 + I_6} \right) \frac{\partial^2 A}{\partial Y^2} + \left( \frac{I_3}{I_1 + I_{63}} \right) \frac{\partial^4 A}{\partial X^4} + \left( \frac{I_5}{I_1 + I_6} \right) A \frac{\partial^2 A}{\partial X^2} = 0 \tag{37}
$$

Rewriting Eq. (37) as

$$
\frac{\partial^2 A}{\partial T^2} + a_1 \frac{\partial^2 A}{\partial T \partial Y} + a_2 \frac{\partial^2 A}{\partial Y^2} + a_3 \frac{\partial^4 A}{\partial X^4} + a_4 \frac{\partial^2 (A^2)}{\partial X^2} = 0. \tag{38}
$$

where

$$
a_1 = \frac{(I_2 - 2c_1 I_6 - c_1 I_1)}{I_1 + I_6} \qquad a_2 = -\frac{c_1 I_2}{I_1}, \qquad a_3 = \frac{I_3}{I_1} \; , \quad a_4 = \frac{I_5}{2 I_1}. \tag{39}
$$

This equation describes the evolution of spatial-temporal amplitude $A(X, Y, \, T)$ of Rossby-
Khantadze waves. When $\ I_2 = 2c_1 I_6 - c_1 I_1 \ $ gives $\ a_1 = 0$, our equation (38) reduces to the



standard Boussinesq equation ((2+1) – dimensional). Otherwise, equation (38) is the general
form of Boussinesq equation (i.e. $a_1 = 0$).

## 5. Dynamical Analysis for the New Boussinesq equation

In order to solve the generalized Boussinesq equation, we follow the methodology
developed by Kaladze et al., (2013b) and later make use of methods of dynamical analysis, to
get extended information about the solution of the equation, and to obtain its trajectories and
fixed points in phase space.
We use the following co-moving frame $A = \emptyset(\xi)$ with $\xi = mX + nY + lT$ to turn Eq.
(39) into an ordinary differential equation. Then after integrating it once over $\xi$ gives us,

$$a_3 m^4 \emptyset'' + (l^2 + a_1 ln + a_2 n^2)\emptyset' + a_4 m^4 \emptyset^2 = g \quad (40)$$
with g as the constant of integration.
We can now express Eq. (40) as a set of two first order autonomous equations as
$$\begin{cases} \dfrac{d\emptyset}{d\xi} = y \, ; \\ \dfrac{dy}{d\xi} = \dfrac{-a_4 m^2 \emptyset^2 - (l^2 + a_1 ln + a_2 n^2)\emptyset + g}{a_3 m^4}. \end{cases} \quad (41)$$

From (40) we express the Hamiltonian of the system as
$$H(\emptyset, y) = \frac{1}{2} y^2 - \frac{a_4 m^2 \emptyset^3}{3 a_3 m^4} - \frac{l^2 + a_1 ln + a_2 n^2}{2 a_3 m^4} \emptyset^2 + \frac{g}{a_3 m^4} \emptyset = h , \quad (42)$$
where $h$ is a constant value.
In order to get the fixed points of our system, we suppose $\left(\dfrac{dy}{d\xi}\right)_{\emptyset_1} = 0$ where $\emptyset_1$ is the fixed
point. Such that,
$$a_4 m^2 \emptyset_1^2 + (l^2 + a_1 ln + a_2 n^2)\emptyset_1 - g = 0. \quad (43)$$
Eq. (43) is a quadratic equation and has two roots, which are given below

$$\emptyset_1 = \frac{-(l^2 + a_1 ln + a_2 n^2)^2 - \sqrt{\Delta}}{2 a_4 m^2}, \quad (44)$$
and

$$\emptyset_2 = \frac{-(l^2 + a_1 ln + a_2 n^2)^2 + \sqrt{\Delta}}{2 a_4 m^2}. \quad (45)$$
where
$$\Delta = (l^2 + a_1 ln + a_2 n^2)^2 + 4 a_4 m^2 g. \quad (46)$$





Let $g_0 = |f(\phi_i) + g|$, then $g_0$ is the extremum values of $f(\phi) + g$.
Suppose $(\phi_i, 0)$ (where $i = 1, 2$) be one of the singular points of the system of equation, then
from our system, the characteristic values
$$\lambda^2(\phi_i, 0) = \frac{f'(\phi_i)}{a_3 a^4}.$$

Based on the qualitative theory for the dynamical system we know that [44]
(i)     If $\frac{f'(\phi_i)}{a^3} < 0$ then $(\phi_i, o)$ is a center point
(i)     If $\frac{f'(\phi_i)}{a^3} > 0$ then $(\phi_i, o)$ is a saddle point
(ii)    If $f'(\phi_i) = 0$ then $(\phi_i, 0)$ is degenerate saddle points
Thus, above analysis provides the bifurcations phase portraits of equation (42).
**5. Solution for the Boussinesq equation**
In this part, based on this dynamical theory, we will deduce the traveling wave solution to
equation (42) by considering $g = 0$.
The equation (41) reduce to the system as follows
$$\begin{cases} \dfrac{d\emptyset}{d\xi} = y, \\ \dfrac{dy}{d\xi} = \dfrac{-a_4 m^2 \emptyset^2 - (l^2 + a_1 l n + a_2 n^2) \emptyset}{a_3 m^4}. \end{cases}$$
(47)

It is expected that equation (41) has a homoclinic orbits $\Gamma_1$.
In $\phi - y$ plane, $\Gamma_1$ is given as
$$y^2 = \frac{2a_4 m^2}{3a_3 m^4} \phi^3 - \frac{(l^2 + a_1 l n + a_2 n^2)}{a_3 m^4} \phi^2,$$
(48)

with $\phi_0 = 3(l^2 + a_1 l n + a_2 n^2)/2a_4 m^4.$
Equations (47) and (48) give

$$\pm \sqrt{\frac{1}{\frac{2a_4}{3a_3 m^2}\phi^3 - \frac{(l^2 + a_1 l n + a_2 n^2)}{a_3 m^4}\phi^2}} \; d\phi = d\xi,$$
(49)

Here we suppose that $\phi(0) = \phi_o$ and integrate (49) along homoclinic orbits $\Gamma_1$, we get

$$\int_\phi^{\phi_o} \frac{ds}{\sqrt{\frac{2a_4}{3a_3\, m^2}s^3 - \frac{(l^2 + a_1 l n + a_2 n^2)}{a_3 m^4}s^2}} = \int_\xi^o ds, \qquad \xi < 0$$
(50)

and



$$\int_{\phi}^{\phi_o} \frac{ds}{\sqrt{\frac{2a_4}{3a_3\, m^2}\, s^3 - \frac{(l^2 + a_1 l\, n + a_2\, n^2)}{a_3 m^4}\, s^2}} = \int_{\xi}^{o} ds, \qquad \xi > 0 \tag{51}$$

Equations (50) and (51) give
$$\phi = \frac{-3\,(l^2 + a_1 l\, n + a_2\, n^2)}{a_4 m^2 [1 - cosh(\eta\xi)]}, \tag{52}$$


$$\phi = \frac{-3\,(l^2 + a_1 l\, n + a_2\, n^2)}{a_4 m^2 [1 + cosh(\eta\xi)]}, \tag{53}$$

where $\eta = \sqrt{\frac{(l^2 + a_1 l\, n + a_2\, n^2)}{a_4 m^4}}$ .
From (52) and (53) along with transformation $A = \phi\,(\xi)$ , $\xi = m\, X + n\, Y + l\, T$   we get
the solution of solitary wave

$$u_1\,(X,\ Y,\ T) = \frac{-3\,(l^2 + a_1 l\, n + a_2\, n^2)}{a_4 m^2 \left[1 - cosh\sqrt{-\frac{(l^2 + a_1 l\, n + a_2\, n^2)}{a_3 m^4}}\zeta\right]}. \tag{54}$$


and
$$u_2\,(X,\ Y,\ T) = \frac{-3(l^2 + a_1 l\, n + a_2\, n^2)}{a_4 m^2 \left[1 + cosh\sqrt{-\frac{(l^2 + a_1 l\, n + a_2\, n^2)}{a_3 m^4}}\zeta\right]}. \tag{55}$$

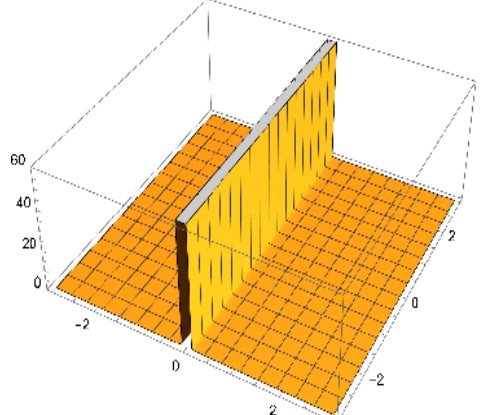


Fig. 2 the solutions (54) are plotted for the parameters $m = n = 1;\ a_1 = a_2 = 0.01;\ a_3 =$
$-0.01;\ a_4 = 10$



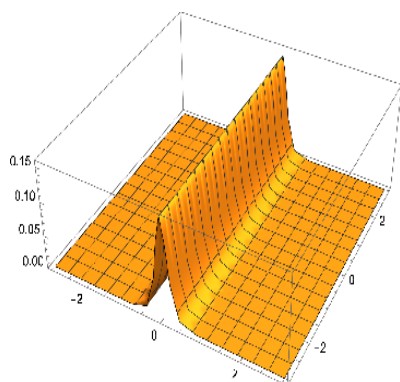


Fig. 3 the solutions (55) are plotted for the parameters $m = n = 1$; $a_1 = a_2 = 0.01$; $a_3 =$

$-0.01$; $a_4 = 10$.

It is shown from the obtained solutions that the considered Rossby-Khantadze waves are
solitary in nature.

6. **Discussion**


In the presented paper, the investigation of large-scale Rossby-Khantadze nonlinear
waves with sheared zonal flows in E-ionosphere plasma is presented. The spatially
inhomogeneous Earth's angular velocity with the background magnetic field are considered.
The spatial inhomogeneity in the field makes possible the coupling of Rossby and Khantadze
waves named Rossby-Khantadze waves (RKWs).
In the work, firstly we considered a system of equations for boussinesq model equation
from the initial set of equations namely, momentum equation, continuity equation and Maxwell
equation telling the nonlinear interaction of considered Rossby-Khantadze waves. By using
curl of our momentum equation we obtain the vorticity equation which is our first system of
equation and in Maxwell equations by taking into account the ionospheric E-region plasma
conditions we got our second system of equation of magnetic induction. Our system of
equations explains how Rossby-Khantadze nonlinear waves propagate in considered sheared
zonal flow ionospheric E region. In earlier work, the authors take into account Rossby waves
while here we take coupled Rossby and Khantadze waves. For the linear consideration, the
linear dispersion relation of the fast (Khantadze) and slow (Rossby) EM wave in the
ionospheric E - region is analyzed with two modes of frequency $\omega_1$ and $\omega_2$. The numerical
work of obtained frequencies is done. The phase velocities depending on wave number is
shown in Figs. 1 - 5 (with red color describes $\omega_1$ while blue ones to $\omega_2$). For small wave vector,
$\omega_1$ approaches to the finite value, while for the $\omega_2$ becomes $-\infty$. For small $\alpha_0$, strong coupling
is shown between two modes. With increasing $\alpha_0$ the Rossby modes approaches to the positive
values, namely at $\alpha_0 = 1$, it approaches to zero and for the values $\alpha_0 > 1$, its phase velocity
approaches to positive value, while the waves with $\omega_2$ always are propagating along the
latitudinally westward. For large wave vector, both modes lose its dispersing property.
In order to investigate the non-linear behavior of coupled RKWs we have used multiple scale
analysis and asymptotic expansion, to derive nonlinear Boussinesq equation with spatial
dependent coefficients. By using the method of multiple scale and hence considering finite
amplitude perturbations, we obtained a new Boussinesq ((2+1) dimensional) equation. We
have also presented the qualitative description of dynamical systems. Thus, based on the ideas
of our work, we can not only obtain the exact traveling wave solutions in the future research,



but can also do the stability analysis, and determine the parameters at which the onset of chaos
takes place. Furthermore, this can help us to understand not only the solitary profiles, but also
the nonlinear periodic wave solutions associated to the Boussinesq equation.
By taking lowest order O ($\varepsilon^{3/2}$) of Eq. (7) we got an eigen-value equation (21). This order
however does not bring information about the amplitude of the Rossby-Khantadze waves.
Thenceforth we use the next order, O ($\varepsilon^2$) of Eq. (7) and obtain non-singular solutions. The
obtained, however, equation still doesn't provide information about the wave amplitude.
The next order of Eq. (7) provides a longitudinal dispersion effect, which competes with a weak
nonlinear effect. This explains that if the perturbation problem has an effective solution, then
the secular term $F$ must be satisfied Eq. (34), otherwise the wave's amplitude would be infinite
and have no significance in practise. By doing some mathematical steps, from next order we
get the nonlinear Boussinesq equation (41). By considering g=0, we also investigate the
dynamical analysis and have done a fixed points analysis analytically. Also, we obtain the
travelling solitary structures shown in Fig. 6-7. The obtained results might be helpful for
understanding the data which is obtained by satellites orbiting the earth in the ionosphere
region.
For the experimental evidence of RK vortical structures in weakly ionospheric region,
the following properties are expected. EM RK perturbations that represents the variation of the
electric field $\boldsymbol{E}_v = \mathbf{v}_{De} \times \boldsymbol{B}_0$, where $\mathbf{v}_{De} = \boldsymbol{E} \times \boldsymbol{B}_0/B_0^2$ is the drift velocity in comparison of
ordinary Rossby waves. Such RK waves propagate latitudinally with the speed of $|c_B| \approx 2 -$
$20\ km/s$. The frequency ($\omega = k_x c_B$) as well as the phase velocity $c_B$ has dependent on charged
particles's number density and is different in day and night. These perturbations are of high
value $(10^4 - 10^{-1})\ s^{-1}$ with wavelengths $\sim 10^3\ km$. RK waves are accompanied by the
strong pulsations of the geomagnetic field 20-80 nT in compared of ordinary Rossby waves.
Note that Khantadze waves were observed at the launching of spacecrafts in middle and
moderate latitudes Burmaka, et al. (2006) and by the world network of ionospheric and
magnetic observations Sharadze, et al. (1988); Sharadze, et al. (1989); Sharadze, (1991);
Alperovich, et al. (2007). The work of Forbes (1996) gives data analyses for describing the
Rossby waves penetration into ionospheric dynamo E-region.
The considered sheared RK waves give insights on large-scale processes and are
observed mainly during magnetic storms as well as sub-storms, artificial explosions,
earthquakes, etc. Hence, for the future experimental work, the theoretical findings of Rossby-
Khantadze electromagnetic type oscillations will provide valuable information.
**AUTHOR DECLARATIONS:**
**Conflict of Interest**
The authors have no conflicts to disclose.
**Data Availability**
The data that support the findings of this study are available within the article.
**Author contributions.** LZK: conceptualization (equal); formal analysis (equal); investigation
(equal); methodology (equal); writing; original draft (equal); supervision (equal); writing -
review and editing (equal). TDK: conceptualization (equal); investigation (equal);
methodology (equal); writing - review and editing (equal). HAS: methodology (equal);
investigation (equal); supervision (equal); writing - review and editing (equal). TZ: formal
analysis (equal); methodology (equal); writing; - original draft (equal). SAB: investigation
(equal); writing– review and editing (equal).



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
