# Peer review of "A Nonlinear Generalized Boussinesq Equation ((2+1)-D) for Rossby-Khantadze Waves Laila Zafar Kahlon1\*, Tamaz David Kaladze2, 3, Hassan Amir Shah1, Taimoor Zaka1, Sved Assad Ul Azeem Bukhari1"

_EGUsphere, 2025_

## Referee Comment (RC1)

**Reviwer's report on paper 'A Nonlinear Generalized Boussinesq Equation ((2+1)-D) for Rossby-Khantadze Waves'**

This paper addresses the role of sheared Rossby-Khantadze (RK) waves in global atmospheric circulation, particularly in the ionospheric E region. The study builds upon prior research (Kaladze et al., 2011-2014) and explores nonlinear processes associated with RK wave dynamics, including their self-organization into solitary dipolar vortices and their potential to generate sheared zonal flows. They are well studied by many scientists, in which fast and slow electromagnetic wave's propagation features were presented in the shear flow driven ionosphere, fast ones exist in the upper layers of the ionosphere, where magnetic filed effecto n the waves enhances, but the slow electromagnetic waves – at comparably lower altitudes due to Coriolis force (Aburjania et al, 2006, JGR). It seems that authors are not familiar of the previous works or deliberately ignore those works.

In this papers the waves are investigated with the novel approach - a system of equations for boussinesq model equation from the initial set of equations namely, momentum equation, continuity equation and Maxwell equation telling the nonlinear interaction of considered multiple-scale analysis and asymptotic expansion to derive the nonlinear Boussinesq equation, providing a systematic approach to understanding wave interactions in the ionospheric E-region, the results are verifying the works Khantadze et al, who was investigating these waves for aa long time, but I couldnot find the reference on his works. Can the authors explain this?

In overal, the method used in this paper and the results are valuable for investigation of these waves in the ionosphere. With improvements in writing, clarity, and interpretation, the paper could provide significant insights into ionospheric wave dynamics.

---

## Author Comment (AC1)

**Title:** A Nonlinear Generalised Boussinesq Equation ((2+1)-D) for Rossby-Khantadze Waves
**Authors:** Laila Zafar Kahlon et al.
**MS No.:** egusphere-2025-123
**MS type:** Regular paper

**Response to Referee 1:**

Dear Reviewer 1,
We sincerely appreciate the time and effort you have taken to evaluate our manuscript (egusphere-2025-123) and provide constructive feedback. By keeping your all comments in our mind, we have made the following changes in our manuscript:

In Pages 1-2, lines 40-51 in the introduction, the following addition "Such slow long-period planetary waves have phase velocities of the order of 1–100 m/s, which is around the…" have been made alongwith some additional references.
Also, the more specific references have been added by keeping in mind the referee comment, namely Khantadze et al. etc.
The writing in many places have been changed to create clarity.
In support of interpretation, we have improved the figures and text.

Best Regards,
Authors

---

## Author Comment (AC2)

**Title:** A Nonlinear Generalised Boussinesq Equation ((2+1)-D) for Rossby-Khantadze Waves
**Authors:** Laila Zafar Kahlon et al.
**MS No.:** egusphere-2025-123
**MS type:** Regular paper

**Response to Referee 2:**
Dear Reviewer 2,

Thank you for your kind suggestions. By keeping in mind your suggestions, the following addition have been made in the manuscript:

The step-wise following changes have been made:
1. The font size of figure labels has been increased, the legend has been added for blue and red curves. The normalization is already added in the text in p#6, lines 218-219.
2. In Figure 2, the font size for tick labels has been increased and also the axes labels have been made.
3. The abstract has been rewritten more concisely.
4. Throughout the manuscript, the English is improved in the text.

Best Regards,
Authors

---

## Author Response (AR1)

**Title:** A Nonlinear Generalised Boussinesq Equation ((2+1)-D) for Rossby-Khantadze Waves
**Authors:** Laila Zafar Kahlon et al.
**MS No.:** egusphere-2025-123
**MS type:** Regular paper

**Response to Editor:**
Dear Respected Editor,

Thank you for your kind response. All the changes have been highlighted in red color. By keeping in mind your suggestions, the following step-wise changes have been made in the manuscript:

1. In page#15, line#425-435, the summary and conclusions section 7 has been added after section 6 Discussion.
2. In the reference list, the location/ position of the year has been homogenized in reference 1, 7, 8, 11-15, 17, 42, 49, 56, 57, 70-72.
   The reference Lü et al., 2016a & 2016b is quoted in the text, only there was a symbolic error which has been removed. Rest of references have been carefully checked again.
3. The English has been improved throughout the manuscript.
4. In page # 1, line # 17-26, the abstract has been rewritten by incorporating the sentences "the qualitative theory of ODEs" is modified by "…by using the qualitative theory of ordinary differential equations (ODEs)…",
5. In page # 1, line# 22-26, in abstract the following sentence "Through which we obtain ..." have been removed and has been rephrased with the following sentence "The obtained numerical results show that the aforementioned solutions of the traveling waves correspond to Rossby-Khantadze solitons."
6. In page # 1, line 34: " of the Zonal flow's existence in atmospheric regions of atmosphere" is changed by "of the presence of zonal flows in various regions of the terrestrial atmosphere"
7. In page #1, line #36, the following sentence "These ULF perturbations …" is replaced by "These ultra low frequency (ULF)…."
8. In page#1, line 46: this sentence "It is discussed that sheared RK electromagnetic vortices in the E ionospheric region (Kaladze et al., 2011; 2012; 2013a, 2013b; 2014a, 2014b)." is changed by "The existence of sheared RK electromagnetic vortices in the E region of Earth's ionosphere is studied thoroughly by Kaladze et al (Kaladze et al., 2011; 2012; 2013a, 2013b; 2014a, 2014b)"
9. In page#1, line 47-49, "……..self-organization of coupled RK waves into solitary dipolar vortices alongwith the possibility….." is replaced with "In which, the authors have not only shown the self-organization of coupled RK waves into dipolar solitary vortices, but also predicted the generation of magnetic field in the system due to the aforementioned waves"
10. In page#2, line 57: the word transfered is replaced by "transferred"
11. In page#, line 60-61: instead to remove the closing bracket at the end of the sentence "work of Bostrom,1992; Lindqvist et al., 1994; Dovner et al. 1994; Qiang et al., 2001)", the sentence is replaced with "…..by *Freja and Viking satellites* (Bostrom, 1992; Lindqvist et al., 1994; Dovner et al. 1994; Qiang et al., 2001)"

12. In page#2, line 63: yes the reference is already present here "the following reference Jian et al., (2009)" for modified Korteweg-de Vries (MKdV) equation on the starting of this sentence

13. In page#2, line 66-67: " Rossby-Khantadze waves having magnetic field " the following sentence "Zonal flow's generation in the ionosphere's E region by Rossby-Khantadze waves having magnetic field …"is replaced by "Zonal flow's and magnetic field's generation in the ionosphere's E region by Rossby-Khantadze waves"

14. In page#2, line 72, the word "E-area" is replaced by "E-region"

15. In page#2, line 72, the word "Forbes, 1996" is replaced by "Forbes (1996)"

16. In page#2, line 73, the comma has been removed here "Vukcevic and Popovic, (2020)"

17. In page#2, line 80, the word "those" is replaced by "these".

18. In page#2, line 84, the word "to studied" is replaced by "for" --> to study. But better rephrase the sentence, because "... investigated .... to study ... " is doubled and not meaningful.

19. In page#2, line 91, the sentence "… to find exact and the explicit " is replaced by "… to find exact  …"

20. In page#2, line 110, the additional fullstop has been removed.

21. In page#3, line 138, the words "provides role" are replaced by "plays their role".

22. In page#4, line 145, the sentence "…s (Pedlosky (1987); Satoh (2004))" are replaced "as explained by Pedlosky (1987) and Satoh (2004)."

23. In page#6, line 213, ". [Kaladze et al., 2011]" is replaced with "(Kaladze et al., 2011)."

24. In page#9, line 269, the second line coefficient $I_2$ is replaced $I_2 - c_1 I_1$

25. In page#13, line 360, "E-ionosphere plasma" is replaced with "ionospheric plasma found in the E-layer"

26. In page#13, line 362, "makes possible" is replaced with "allows"

27. In page#13, line 367, "By using curl" is replaced with "By taking the curl"

28. In page#13, line 368, the sentence is rephrased with "… by using the Maxwell's

29. electromagnetic equation, by taking the parameters of the E layer of ionosphere into account."

30. In page#, line 363, the abbreviation "(RKWs)" is removed.

31. In page#13, line 373, the word "electromagnetic" is added before (EM)

32. In page#13, line 387, the word "can not"is written without space "cannot"

33. In page#13, the first letter of  Sun and Earth is capitalized throughout the manuscript.

34. In page#14, lines 404-416: the whole paragraph is rephrased and added in introduction section.

35. In page#, line 413: the bracket Burmaka, et al. (2006) has been changed with "(Burmaka et al. 2006)" the same changes have been made in remaining references of these sentences.

Some additional changes have been made:
1. The whole abstract has been improved.
2. New references 31-33, 52, 54 have been added.
3. In page#1, lines 38-39, the sentence "The sheared flow affects …." is changed by "The effects of sheared flow…….".
4. In page#1, line 41, the word ",and" has been added.
5. In page#1, lines 43-44, the sentence "It must be noted that the spatial inhomogeneity, along …" is replaced by "The spatial inhomogeneity along the meridians, of both …"

6. In page#1, line 46, the following lines "Such slow long-period planetary waves have phase velocities ~ 1–100 m/s, which …" have been added.
7. In page#1, lines 47-48, the following sentence "In the aforementioned papers, the …" is changed with "In those works, the authors …"
8. In page#1, line 49, the words "In the recent work, …" is changed with "More recently, …"
9. In page#2, lines 52-53, the sentence "The Rossby waves …" is changed with "Rossby waves causes the generation of zonal flows in E layer of ionosphere ….., …"
10. In page#2, line 54, the words "are splitted into various parts having dependent on zonal…" is changed with "splits into various parts, and this splitting is dependent on …"
11. In page#2, line 58, the sentence "While the equatorially propagating Rossby…" is changed with "It is worth noting that …"
12. In page#2, lines 67-68, the words "having magnetic field have …). It has been …" is changed with "is also shown …), where it …"
13. In page#2, lines 70-71, the words "… in which they obtained MKdV solitons. …" is changed with "in which MKdV solitons were obtained."
14. In page#2, lines 78-79, the following sentence "Notably, Kadomstev-Petviashvilli (KP) equation and Zakharov-Kuznetsov (ZK) equation have gained much attention" is replaced "Kadomstev-Petviashvilli (KP) equation and Zakharov-Kuznetsov (ZK) equation have gained much attention over the years …".
15. In page#2, lines 78-79, the word "those" should replaced with "these"
16. In page#2, line 91, the words "to find exact and the explicit nonlinear solutions of partial differential" are replaced with " … to find exact solutions of nonlinear partial differential"
17. In page#3, line 105, the names is replaced twice with Lü.
18. In page#3, lines 106-107, the words "to have some of the techniques." is replaced with "to name a few."
19. In page#3, line 115, the following sentence "The summary and conclusion are made in Sec. 7." has been added.
20. In page#3, line 126, the word "along" has been added.
21. In page#3, lines 127, the word "Here" has been added.
22. In page#3, lines 127, the word "as" has been added.
23. In page#3, lines 134, the word "waves is expressed" has been added.
24. In page#3, lines 134, the word ", is" is replaced with ", being".
25. In page#3, lines 134, the words "provides role" is replaced with "plays their role".
26. In page#3, line 145, the words "directions (Pedlosky (1987); Satoh (2004))." Are replaced with ", as explained by Pedlosky (1987) and Satoh (2004)."
27. In page#3, line 173, "waves, we will use the multiple …" Is replaced with ". Here we make use of multiple …"
28. In page#4, line 161-163, the words "The stream function is taken as….." is replaced with "The expression….."
29. In page#4, line 171, the following sentence "approach we find the following stretched coordinates," has been added.
30. In page#6, lines 214-216, the lines "In Fig. 1, the phase velocity … . Red curve is for "+" and …." are replaced by "Fig. 1, represents … of the obtained … . Red curve $v_{p1}$ is for "+" and blue $v_{p2}$ is for "–" signs before the radicand in Eq. (21)."
31. In page#7, lines 225-226, in the caption the word "Normalized…… is shown." has been added.

32. In page#7, line 235, the following line "… Eq. (22) and (23) in Eq. (13) we obtain" has been added.
33. In page#9, line 273, the words "which gives" are replaced with "we obtain".
34. In page#11, line 324, the following line (which corresponds to a solitary wave profile) has been added.
35. In page#13, lines 366-369, the lines have been rephrased, "By taking the curl of momentum … ionosphere into account."
36. In page#13, lines 373, the word "electromagnetic" has been added.
37. In page#13, lines 387, the word "can not" is written as "cannot".
38. In page#14, lines 401, the words "Figs. 6-7" is replaced with "Figs. 2-3".
39. In page#14, lines 404-416, these lines are removed.

Best Regards,
Authors